# Optimizing nitrogen removal with an immobilized biological filler system: realizing stage-independent operational process

Xuyan Liu[1,2☯], Hong Yang[3], Jiawei Wang [ID][4]*

1 Hebei GEO University, Shijiazhuang, China, 2 Hebei Center for Ecological and Environmental Geology Research, Hebei GEO University, Shijiazhuang, China, 3 Key Laboratory of Beijing Water Quality Science and Water Environment Recovery Engineering, Beijing University of Technology, Chaoyang, Beijing, China, 4 Hebei Key Laboratory of Water Quality Engineering and Comprehensive Utilization of Water Resources, Hebei University of Architecture, Zhangjiakou, China

☯ These author contributed equally to this work.
* ykaisjz@163.com

## Abstract

Due to the operation mode of traditional activated sludge systems, it is difficult for various functional bacteria to exert their respective advantages. In this study, immobilized fillers for hydrolytic acidification, nitrification, and denitrification were developed to allow independent operation at each stage, enhancing nitrogen removal performance of overall process. The results showed that ammonia nitrogen and total nitrogen levels in the effluent stabilized at 0.75–0.83 and 1.5–2 mg/L, respectively, when the total hydraulic retention time (HRT) of the system was 6.4 h and the nitrification unit HRT was 3 h. These values represented significant improvements compared with the traditional activated sludge process. Unit performance tests revealed that reducing the hydrolytic-acidification time to 0 min increased nitrite nitrogen and nitrate nitrogen levels in the effluent of unit A2 to 6.11 ± 0.2 mg/L and 3.67 ± 0.1 mg/L, respectively. This demonstrates that an active hydrolysis - acidification stage is the prerequisite for A2 to fully utilize raw organic matter in the water for remove nitrogen. When raw organic matter in the water bypassed the A2 unit and entered the O1 unit directly, ammonia oxidation rate (AOR) significantly decreased (from 0.32–0.33 to 0.22–0.23 kg/m³·d), with further reduction at a low temperature (down to 0.11–0.12 kg/m³·d). At this time, the AOR, unaffected by organic matter, decreased only slightly. This indicates that directing organic matter into the nitrification stage is essential for maintaining stability and resisting low temperatures. This process has certain guiding significance for improving nitrogen removal efficiency in municipal wastewater processes.

## Introduction

Municipal sewage treatment plants use activated sludge to remove nitrogen and phosphorus from wastewater using bacterial hydrolytic acidification along with nitrifying, denitrifying, and polyphosphate-accumulating bacteria. In a single sludge system, pollutants are removed through three treatment units (anaerobic, anoxic, and aerobic). The processed water enters the sedimentation tank for solid-liquid separation and bacterial collection; it then flows back

**Data availability statement:** All relevant data are within the manuscript and its Supporting Information files.

**Funding:** Xuyan Liu:Science Research Project of Hebei Education Department (QN2024036); Xuyan Liu:the Open Project Program of Hebei Center for Ecological and Environmental Geology Research (JSYF-202304); Hong Yang: the Beijing Municipal Commission of Education (Z161100004516015); Xuyan Liu: The Doctoral Research Start-up Fund of Hebei GEO University(BQ2024084). The sponsor or funder plays a role in research design, data collection and analysis, publication decisions or manuscript preparation.

**Competing interests:** The authors have declared that no competing interests exist.

**Abbreviations:** TN, Total nitrogen; NH4 + -N, Ammonia nitrogen; NO2—N, Nitrite nitrogen; NO3—N, Nitrate nitrogen; COD, Chemical oxygen demand; DON, Dissolved organic nitrogen; DO, Dissolved oxygen; HRT, Hydraulic retention time; AOR, Ammonia oxidation rate; ARR, Ammonia nitrogen removal rate; NRR, TN removal rate; SEM, Scanning electron microscopy; VFAs, Volatile fatty acids.

to the front end of the tank to retrieve biomass [1]. In this method, bacteria with different physiological characteristics and generation cycles function in the same system [2], which makes it difficult to leverage their respective advantages and results in lower overall treatment efficiency.

In the traditional activated sludge process, sludge reflux to maintain the biomass in the reaction system is generally employed, which inhibits the formation and maintenance of a strict anaerobic environment for hydrolysis-acidification bacteria. This makes it difficult to fully transform complex organic matter into simple organic matter and requires the addition of a carbon source. Furthermore, complex organic matter needs to be oxidized in the aerobic stage, which results in ineffective energy consumption. In addition, the influent dissolved organic nitrogen (DON) cannot be fully converted into $NH_4^+$-N and removed, affecting overall treatment efficiency [3–5].

The biological effect of the traditional activated sludge method is highly contradictory. The specific operation mode results in a mixture of various functional bacteria, and the performance of those involved in nitrification and denitrification for nitrogen removal cannot be further improved [6,7]. In addition, since nitrifying bacteria have the longest reproduction cycle, longer hydraulic retention times (HRTs) and a higher sludge age (SRT) are required to ensure nitrification[8,9]; however, bacteria with a short reproduction cycle, such as phosphorus-accumulating bacteria, need to be continuously discharged to achieve good treatment results. This discharge decreases the number of nitrifying bacteria in the reactor, resulting in a decrease in overall nitrogen removal performance [10–12].

It is necessary to increase the overall sludge concentration or provide suitable conditions in traditional activated sludge treatment to better leverage the characteristics of certain bacteria. However, a high sludge concentration causes sludge swelling and loss. Attachment culturing is generally performed to obtain biofilms to achieve the separation of functional bacteria. However, owing to the inability to selectively control the types of attached bacteria, it is difficult to effectively separate various types of functional groups [13]. Immobilization technology has received widespread attention, as it facilitates the targeted cultivation of specific spatial areas through physical or chemical means, stabilizing the bacterial population while maintaining high activity and controlling species separation [14,15].

This study utilized urban sewage and immobilized hydrolytic acidification, nitrification, and denitrification fillers to build an anaerobic-anoxic-aerobic anoxic (AAOA) pilot nitrogen removal system. This system enabled the separation of different functional bacteria to fully utilize their respective properties and also solved the problems of the poor rate of influent organic matter utilization and low nitrogen removal efficiency seen in the traditional activated sludge process. The objectives were to determine the overall operation performance of the AAOA nitrogen removal system under different working conditions and seasonal temperature changes, along with the key control parameters, characterize each unit (including the influence of hydrolytic acidification on first-order denitrification performance and organic matter on nitrification performance), and analyze the internal structure of each filler. These results provide a basis for the efficient and stable removal of nitrogen from municipal sewage.

## Materials and methods

### Experimental water quality

The influent of the AAOA nitrogen removal pilot system originated from the primary sedimentation tank of the Beijing Gaobeidian Sewage Treatment Plant (Beijing, China). Table 1 provides an overview of the water quality of the influent.

**Table 1. Water quality characteristics of the municipal wastewater.**

| Items | Unit | Range | Average |
|---|---|---|---|
| COD[a] | mg/L | 103.4–156.8 | 132.5 |
| TN[b] | mg/L | 42.4–50.3 | 45.2 |
| $NH_4^+$-N[c] | mg/L | 37.6–44.3 | 40.2 |
| $NO_2^-$-N[d] | mg/L | 0.2–0.4 | 0.3 |
| $NO_3^-$-N[e] | mg/L | 0.6–0.9 | 0.8 |
| VFA[f] | mg/L | 75.4–80.4 | 78.8 |
| Organic nitrogen | mg/L | 2.3–10.4 | 6.7 |
| pH | – | 7.6–8.1 | 7.9 |

[a]COD: Chemical oxygen demand;

[b]TN: Total nitrogen;

[c]$NH_4^+$-N: Ammonia nitrogen;

[d]$NO_2^-$-N: Nitrite nitrogen;

[e]$NO_3^-$-N: Nitrate nitrogen;

[f]VFAs: Volatile fatty acids

## Preparation of the immobilized filler

Polyvinyl alcohol was weighed and dissolved in water. It was then mixed with different bacteria, and auxiliary additives were added to form a relatively viscous embedded solution. The different bacterial auxiliary agents were added in different ways. Equipment was developed in the laboratory to pressure the concentrated mixed liquid to form a hollow cylinder that was cut into small sections. These sections were then placed in a solution of saturated boric acid for crosslinking and stabilization. Finally, they were cleaned with water to prepare immobilized fillers for hydrolytic acidification, nitrification, and denitrification. The filler had no skeleton and was a hollow cylinder approximately 1 cm wide and 2 mm thick [16–18].

## Reactor setup and operation

The anaerobic-anoxic-aerobic-anoxic (AAOA) immobilized filler nitrogen removal system consisted of hydrolytic acidification, nitrification, and denitrification fillers that were components of a hydrolytic acidification unit (A1; effective volume 200 L), a primary denitrification unit (A2; effective volume 320 L), a nitrification unit (O1-O3; effective volume 200 L×3), and a secondary denitrification unit (A3; effective volume 160 L); the total volume was 1.28 m³ (Fig 1). Influent water first entered unit A1 for hydrolysis and acidification to transform the complex organic matter into easily degradable organic matter and fully transform the DON. It subsequently entered unit A2 for denitrification by fully using influent COD and $NO_3^-$-N. It then entered unit O, which was divided into three parts for gradient aeration; in this manner, $NH_4^+$-N could be converted into $NO_3^-$-N while controlling the DO that entered the denitrification compartment. Finally, it entered unit A3, and the remaining $NO_3^-$-N was converted into $N_2$.

## Impacts of the working conditions on AAOA operation

The operating parameters of the system are shown in Table 2. The first phase was the stable operation phase, which was divided into three parts: the reflux ratio optimization stage (1 to 40 d), the HRT exploration stage (41 to 60 d), and the operation stage under seasonal temperature changes (61 to 120 d). The effects of reflux ratio, HRT, and seasonal temperature

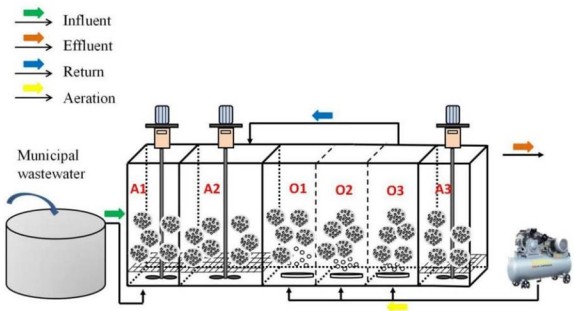

**Fig 1. Schematic diagram of the anaerobic - anoxic - aerobic - anoxic (AAOA) nitrogen removal process.**

**Table 2. Parameters of the operation of the AAOA immobilized filler nitrogen removal system.**

| | Day(d) | Temperature(°C) | A1 | A2 | O1-O3 | A3 | Reflux ratio |
|---|---|---|---|---|---|---|---|
| | | | HRT(h)[a] | HRT(h)[a] | HRT(h)[a] | HRT(h)[a] | % |
| Phase I | 1–10 | 24±1 | 1 | 1.6 | 3 | 0.8 | 80 |
| | 11–20(P1) | | | | | | 90 |
| | 21–30 | | | | | | 100 |
| | 31–40 | | | | | | 110 |
| | 41–50(P2) | | 0.75 | 1.2 | 2.27 | 0.6 | 90 |
| | 51–60(P3) | | 0.5 | 0.8 | 1.5 | 0.4 | |
| | 61–100 | 23±1~17±1 | 1 | 1.6 | 3 | 0.8 | |
| | 101–120 | 17±1~14±1 | 1.33 | 2.12 | 4 | 1.06 | |
| Phase II | 121–160 | 13±1 | 1.67 | 2.67 | 5 | 1.34 | 100 |

[a]HRT: Hydraulic retention time

on the nitrogen removal of the AAOA fillers were investigated. The second phase was the low-temperature operation phase (121 to 160 d). The low-temperature operation of municipal sewage treatment is a bottleneck that needs to be solved, so this phase was investigated to improve the efficiency of this process.

### Effect of organic matter on nitrification performance

The primary purpose of this study was to investigate the effect of organic matter on nitrification. By removing the first-stage denitrification (A2) unit, the influent water directly entered the O unit after passing through the A1 unit. This enabled a comparison to study the effect of organic matter on the performance of the nitrification unit. The schematic diagram of the process is shown in Fig 2, and the specific operation parameters are shown in Table 3.

### Equations and calculations

The ammonia oxidation rate (AOR) and the ammonia nitrogen removal rate (ARR) were calculated using Equations 1, 2, and 3:

$$\text{AOR1}\left(\text{kg/m}^3\cdot\text{d}\right) = \frac{\left(\left(\text{NH}_4^+ - \text{N}\right)_{\text{inf}} - \left(\text{NH}_4^+ - \text{N}\right)_{\text{eff}}\right)\times 24}{t_{\text{HRT}}\times 1000} \tag{1}$$

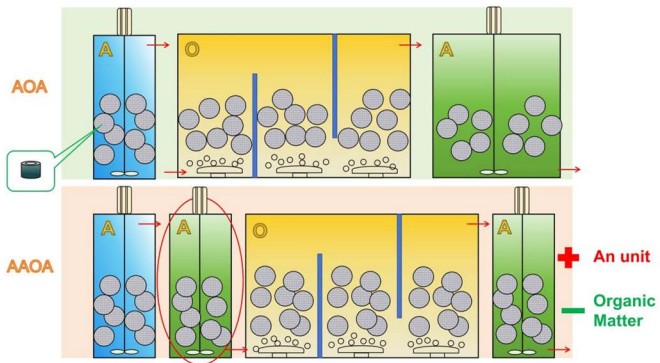

**Fig 2. Schematic diagram of the process changes under the influence of organic matter, including the AOA and AAOA nitrogen removal processes.**

**Table 3. Effects of operating parameters and organic matter on the performance of the nitrification unit.**

|  | Days (ᵈ) | Temperature (°C) | A1 unit | O unit | A3 unit |
|---|---|---|---|---|---|
|  |  |  | HRT (h) | HRT (h) | HRT (h) |
| Phase I | 1–50 | 23 ± 1–17 ± 1 | 0.75 | 2.27 | 1.2 |
| PhaseII | 51–70 | 17 ± 1–14 ± 1 |  |  |  |
|  | 71–110 | 13 ± 1 | 1.33–1.67 | 4–5 | 2.12–2.67 |

$$AOR2 \left(kg/m^3 \cdot d\right) = \frac{\left(Q_{inf} \times \left(NH_4^+ - N\right)_{inf} - \left(NH_4^+ - N\right)_{eff} \times \left(Q_{eff} + Q_{ref}\right)\right)}{V} \quad (2)$$

$$ARR(\%) = \frac{\left(NH_4^+ - N\right)_{inf} - \left(NH_4^+ - N\right)_{eff}}{\left(NH_4^+ - N\right)_{inf}} \times 100 \quad (3)$$

where $(NH_4^+\text{-}N)_{inf}$ is the influent $NH_4^+$-N concentration; $(NH_4^+\text{-}N)_{eff}$ is the influent $NH_4^+$-N concentration, and $t_{HRT}$ is the hydraulic retention time.

The TN removal rate (NRR) was calculated using Equation 4:

$$NRR(\%) = \frac{(TN)_{inf} - (TN)_{eff}}{(TN)_{inf}} \times 100 \quad (4)$$

Where $(TN)_{inf}$ is the influent TN concentration, and $(TN)_{eff}$ is the influent TN concentration.

## The distribution of TN in each part of the process was calculated as follows:

$$Mass\left(Influent \ TN\right) = Q \times TN_{inf} \quad (5)$$

The daily N inflow was calculated from the inflow and the TN concentration, including DON, $NH_4^+$-N, and $NO_3^-$-N; the $NO_2^-$-N concentration was negligible.

$$TN\left(Mass\left(Influent \ TN\right)\right) = MassTN_{A2,A3} + MassTN_O + Mass\left(Effluent \ TN\right) \quad (6)$$

In the AAOA system, the N was removed through primary and secondary denitrification, and the TN amount in this stage was Mass $TN_{A2,A3}$. Aerobic denitrification occurred in the nitrification stage, where TN was Mass TNo. Since the N removal system was composed of immobilized fillers and did not consider the residual sludge problem, the residual TN was taken as the final effluent N content. The formula for this calculation is as follows:

$$\text{Mass}(\text{Efflunet TN}) = Q \times TN_{eff} \tag{7}$$

The daily effluent N level was calculated from the effluent flow and the TN concentration.

## Analytical methods

The levels of $NH_4^+$-N, $NO_2^-$-N, $NO_3^-$-N, and TN were determined using the national standard method (SEPAC, 2002) with Nessler reagent spectrophotometry, N-(1-naphthyl)-ethylenediamine spectrophotometry, ultraviolet spectrophotometry (UV-1600PC, MAPADA), and potassium persulfate oxidation ultraviolet spectrophotometry, respectively [19–21]. The COD level was determined using a rapid COD measuring instrument (Lianhua Technology, Shenzhen, China)[22].

## Scanning electron microscopy (SEM)

We performed SEM to observe and analyze the internal structure of the immobilized fillers of each strain. Different immobilized fillers were obtained and immersed in the sampling tube with 2.5% glutaraldehyde, and the pH was adjusted to 6.8. The samples were fixed at 4 °C for 1.5 h and rinsed two to three times with 0.1 mol/L phosphoric acid buffer for 10 min each. The samples were then dehydrated with 50%, 70%, 80%, and 90% ethanol for 10–15 min in sequence and subsequently dehydrated three times with 100% ethanol for 10 to 15 min. The samples were replaced by a 1:1 mixture of ethanol and isoamyl acetate (v/v) and a pure solution of isoamyl acetate. Each incubation time was within 10–15 min. After replacement, the samples were placed in the dryer for > 8 h, and those with poor electrical conductivity were sprayed with gold. Finally, the treated samples were placed under the SEM, and the different positions and magnifications were adjusted according to the specific requirements. In all, 10–20 clear structural diagrams were obtained for analysis [23].

## Community structure analysis

Bacterial 16S rRNA high-throughput sequencing was used to investigate the microbial community diversity of sludge. Illumina HiSeq 2500 PE250 (Illumina, San Diego, CA, USA) was used to amplify and sequence the V3–V4 region of bacterial 16 S rRNA, and Megan software was used to perform the environmental 16S sequence analysis. High-quality sequences of the corresponding samples were obtained and divided into different clustering operational taxonomic units (OTUs). Finally, the OTU results were analyzed and calculated [24,25].

## Results and discussion

### Impacts of working conditions on the operation of the AAOA system

**Impact of reflux ratio.** We tested to select the most suitable internal reflux ratio (R) (80%, 90%, 100%, and 110%) over 40 d of operation. The apparent $NH_4^+$-N value of each unit of the system was diluted by reflux. On days 1 to 10, when R was 80%, the concentration of $NH_4^+$-N in the effluent of unit A2 ranged from 23.16 to 24.89 mg/L; simultaneously, the concentration of $NH_4^+$-N in the effluent of the O1 unit decreased to 15.03–16.56 mg/L.

According to Equation 2, the AOR value at this time was between 0.25 and 0.26 kg/m³·d, and the ARR ranged from 32% to 34%. The concentration of $NH_4^+$-N in the effluent water of the O2 unit was further reduced to 5.75–6.33 mg/L; the AOR value was 0.42–0.44 kg/m³·d, and the ARR was 61.7–64.6%. After entering the O3 treatment unit, the concentration of $NH_4^+$-N decreased to 0.82–0.97 mg/L; the AOR value was 0.21–0.23 kg/m³·d, and the ARR was as high as 83.7–86.5%. In addition, the effluent COD concentration of the A2 treatment unit was 64.4–70.5 mg/L.

When R increased to 90% (11–20 d), the $NH_4^+$-N effluent of O1 decreased to 13.38–13.5 mg/L and the AOR increased to 0.33–0.35 kg/m³·d compared to those seen with R at 80%. At this time, the COD of the A2 effluent was 45.5–48.3 mg/L, which indicated that the COD had an impact on the AOR of O1. When R was 90%, A2 had sufficient $NO_3^-$-N for denitrification and largely consumed the influent COD. This impeded the growth of the heterotrophic bacteria that were introduced into the system through the influent because they could not fully use the COD during the aerobic stage, meaning they could not compete with the nitrifying bacteria for dissolved oxygen [26,27]. At this time, there was little change in the ARR and AOR of O2 and O3, and the $NH_4^+$-N in the effluent remained below 1 mg/L. When R increased to 100% (d 21–30), the COD of the A2 effluent was 45.67–47.89 mg/L, which was not significantly different from the previous stage, indicating that there was a negative effect on the degradation of organic matter. With the degradation effect in the aerobic stage, the COD of the O3 effluent was reduced to 31.67–33.19 mg/L, whereas the ARR of O1 in this stage remained at 0.33–0.35 kg/m³·d. Similarly, when R increased to 110% (31–40 d), the ARR of O1 remained unchanged, and the $NH_4^+$-N level in the effluent from both stages remained below 1 mg/L. This indicated that when R was 90%, the $NO_3^-$-N reflux generated by the nitrification unit was sufficient for A2 to consume most of the easily degradable COD in the influent water. However, an additional increase of R did not considerably improve the performance of the entire nitrification unit but increased its energy consumption [28–30]. From these results, an R-value of 90% was deemed the most suitable.

The changes in TN of each unit in the AAOA system are shown in Fig 3(c). From days 1 to 10, when R was 80%, there was less $NO_2^-$-N and $NO_3^-$-N in the A2 effluent compared with traditional process, and the TN in the O1 effluent decreased by approximately 3.5 mg/L compared to A2. The NRR was 13.75%–14.33%. The NRR of O2 was 9.49%–9.96%, whereas the NRR of O3 was only 3%–4%. The TN of the A3 effluent was 1.59–2.05 mg/L. When R increased to 90%, the $NO_2^-$-N and $NO_3^-$-N levels in the A2 effluent increased by 0.68–0.8 and 0.98–1.03 mg/L, respectively. At this time, the NRR of O1 was reduced to 4.56%–4.69%; the NRR of O2 and O3 was only 3%–4%, and the TN of the A3 effluent was 1.5–2 mg/L. These findings indicate that when R ranged from 90% to 80%, the easily degradable organic matter in the influent was fully utilized. This not only prevented the nitrification unit from being affected by the presence of COD but also reduced the effect of aerobic denitrification, resulting in a decrease in the variation of effluent TN in each unit [31,32]. When R increased to 100% and 110%, the $NO_2^-$-N and $NO_3^-$-N levels in the A2 effluent increased slightly, which was related to insufficient levels of CODs in the influent. For the nitrification unit, the consumption of COD resulted in a small variation in effluent TN, and owing to the increase in R, the inlet load of A3 was reduced, and the effluent TN could be maintained at a low level [33,34].

**Impact of HRT.** The effect of HRT on the system was also assessed. The first stage (P1) occurred from days 11 to 21, with a total HRT of 6.4 h and a nitrification unit HRT of 3 h. The second stage (P2) lasted from days 41 to 50; the total HRT was 4.82 h, and the nitrification unit HRT was 2.27 h. The third stage (P3) was from days 51 to 60, with a total HRT of 3.2 h and a nitrification unit HRT of 1.5 h. The specific operation parameters are shown in Table 2.

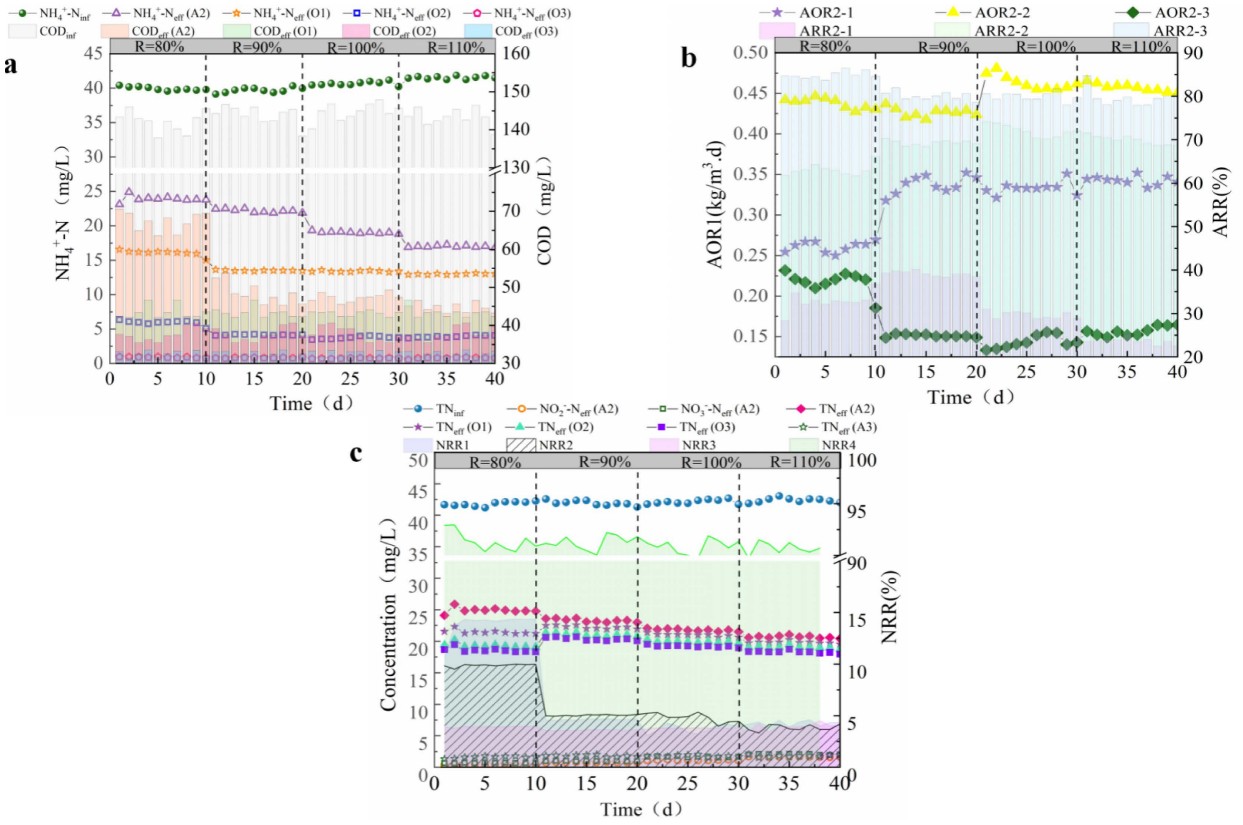

**Fig 3. AAOA system optimized reflux ratio stage performance.** (a) Changes in NH$_4^+$-N. (b) AOR and ARR. (c) Changes in TN ((AAOA: Anaerobic - anoxic - aerobic - anoxic; NH$_4^+$-N: Ammonia nitrogen; AOR: Ammonia oxidation rate; ARR: Ammonia removal rate).

The influence of HRT on the operation of the nitrification unit is shown in Fig. 4(a,b). During the P1 stage, the nitrification units performed well, with an AOR of 0.27–0.28, 0.47–0.48, and 0.16–0.18 kg/m³·d and an ARR of 39%–40%, 69%–70%, and 79%–80%, respectively. The effluent NH$_4^+$-N of the nitrification unit stabilized at 0.78 mg/L. When the HRT of the nitrification unit was reduced to 2.27 h, and the reaction time of each unit was 0.75 h, the nitrification unit performed well and even exceeded the performance of the previous stage at certain points (days 46–48). The COD of each unit's effluent increased compared to that of the previous stage: that of O1 increased from 47.3 to 53.4 mg/L, that of O2 increased from 43.3 to 49.8 mg/L, and that of O3 increased from 36.2 to 44.9 mg/L. When the HRT of the nitrification unit was reduced to 2.27 h, the performance did not decrease but the concentration of COD in the effluent increased, which indicated that aerobic denitrification was easily affected by HRT [34,35]. When the HRT was further reduced to 1.5 h, the effluent NH$_4^+$-N of the nitrification unit increased greatly and reached 3.89 mg/L on day 60. The AOR decreased to 0.12–0.13, 0.43–0.44, and 0.17–0.18 kg/m³·d, and the ARR decreased to 31.4%–32.7%, 52.5%–53.4%, and 45.3%–46.3%, respectively. This indicated that there was insufficient time for the reaction to proceed.

At an HRT of 6.4 h, the units of the AAOA system performed well (Fig 4 (c)). There were low concentrations of NO$_2^-$-N and NO$_3^-$-N in the A2 effluent, and the NRR of the nitrification units was between 4.5% and 5.0%. The TN in the A3 effluent ranged between 1.76 and 1.89 mg/L, and the NRR was between 90.79% and 91.92%. The performance remained stable when the HRT was decreased to 4.82 h. There were low concentrations of NO$_2^-$-N and NO$_3^-$-N

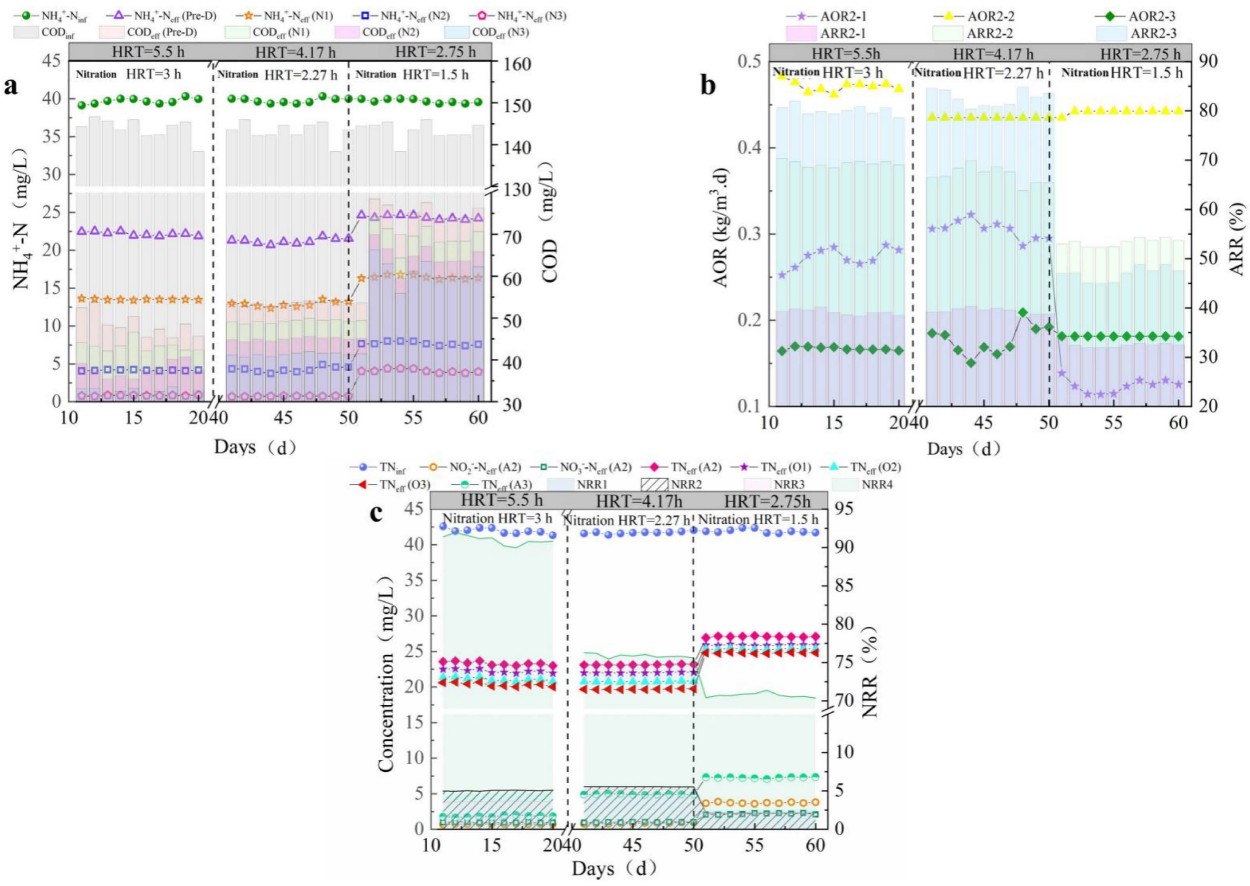

**Fig 4. Optimization of the HRT stage performance in the AAOA system.** (a) Changes in **NH4+-N concentration; (b) AOR and ARR; (c) changes in TN (AAOA: a**naerobic - anoxic - aerobic - anoxic; **HRT: h**ydraulic **r**etention **t**ime; **NH4+-N: a**mmonia **n**itrogen; **AOR: a**mmonia **oxidation rate; ARR: a**mmonia **removal rate).**

in the effluent of A3. However, the residual TN was 4.83–5.01 mg/L, and the NRR decreased to 75.6%–76.2%. When the HRT decreased to 3.2 h, $NO_2^-$-N accumulated in the A2 unit and the NRR in the nitrification unit decreased to 2.09%–2.56%, indicating that aerobic denitrification had diminished. The TN that remained in the effluent of A3 was 7.0–7.3 mg/L. Combined with the previous analysis of the changes of $NH_4^+$-N, the $NH_4^+$-N concentration at this time was found to include DON transformation. The residual $NO_3^-$-N concentration of the denitrification unit was approximately 3.5–3.6 mg/L.

Based on these results, each unit maintained a good state at an HRT of 6.4 h. When the HRT decreased to 4.82 h, the HRT of the nitrification unit was 2.27 h. At this time, the nitrification unit still maintained high levels of AOR and ARR, but there was residual TN in the effluent of A3. The concentrations of $NO_2^-$-N and $NO_3^-$-N remained low, which indicated that the reaction time of A1 was too short. This resulted in residual DON and an increase in effluent TN. As the HRT further decreased to 3.2 h, the nitrification stage HRT was only 1.5 h. At this time, although $NH_4^+$-N and TN residues appeared in the effluent, the national Class A standard was still met. This indicated that the fillers in the AAOA system facilitated efficient nitrification and denitrification. However, to ensure a stable and efficient operation of the system, a total HRT of 6.4 h and a nitrification unit HRT of 3 h were selected in the subsequent stages to maximize the performance of each unit [36].

**Impact of temperature.** In this study, the operational performance of the system under seasonal temperature changes and low temperatures in winter was explored (61 to 120 d). The specific operating parameters are shown in Table 2.

The temperature gradually decreased from days 61 to 100. In the early stage of seasonal temperature changes, the performance of each nitrification unit did not change considerably, and the AOR of O1, O2, and O3 remained at 0.32–0.35, 0.46–0.48, and 0.15–0.17 kg/$m^3$·d, respectively. The levels of ARR1, ARR2, and ARR3 reached 38%–39%, 69%–70%, and 79%–80%, respectively, and the effluent $NH_4^+$-N remained below 1 mg/L. When the temperature decreased to 17 °C, the effluent $NH_4^+$-N level started to increase; the AOR of O1 and O2 decreased to 0.28–0.29 and 0.34–0.35 kg/$m^3$·d, respectively, and the AOR of O3 increased to 0.18–0.19 kg/$m^3$·d. These changes were related to the increase in the effluent load of the previous unit owing to the lower temperature. The ARR decreased to 34%–35%, 47%–48%, and 51%–52%, respectively, and the effluent level of the nitrification unit increased to 3.67 mg/L, indicating that when the temperature reached 17 °C, it negatively affected the performance of the filler. When the temperature decreased to < 14 °C, the level of effluent in O3 increased to 3.96 mg/L. At this time, the system entered the low-temperature operation stage; the AOR was stabilized at 0.32–0.33, 0.33–0.34, and 0.18–0.19 kg/$m^3$·d, respectively, and the effluent $NH_4^+$-N decreased to 3.42 mg/L and remained stable (Fig 5). This indicated that the nitrifying fillers in the AAOA system could maintain high activity when experiencing seasonal temperature changes, though system performance was affected to some extent at temperatures below 17 °C, and temperatures below 14 °C substantially affected performance. However, by adjusting the operating conditions, the effluent levels still met the national Class A standard [37].

When the temperature decreased seasonally, the NRR of each unit remained stable, and there were low effluent nitrogen levels in A2 (Fig 5(c)): the level of $NO_2^-$-N was 0.73–0.75 mg/L, that of $NO_3^-$-N was stable at 0.95–1.01 mg/L, and the effluent TN was 22.9–23.1 mg/L. After the O unit reaction, the NRR1 was 4.7%, whereas the NRR2 and NRR3 levels were 5.1% and 3.6%, respectively. These values indicated that aerobic denitrification occurred in the nitrification unit. The final effluent TN of A3 remained at approximately 1.5 mg/L, and the NRR4 was 94%. When the temperature decreased to 14.2 °C (14 d), the TN in the A3 effluent began to increase; when the temperature dropped below 13.6 °C, the increase was 5.03 mg/L, and the NRR4 was 70.3%–70.5%. However, the NRR of the nitrification units increased in the low-temperature operation stage because the effect of aerobic denitrification was enhanced after the adjustment of the operating conditions; the extended HRT was 4.5–5 h. These results indicate that the denitrification filler was strongly resistant to low temperatures and could maintain stable treatment; although the performance decreased when the temperature decreased to 14 °C or lower, the Class A requirements were still met.

## Nitrogen changes along the AAOA system

After the action of A1, the mean value of $NH_4^+$-N increased from 35.67 to 40.66 mg/L, whereas the levels of TN, $NO_2^-$-N, and $NO_3^-$-N changed only slightly, and the COD decreased from 139.2 to 125.4 mg/L. This indicated that the DON conversion was approximately 5 mg/L in the hydrolytic acidification stage. After the action of the A2 unit, the effluent TN remained at 24.66 mg/L and the COD at 46.77 mg/L. After the nitrification unit reaction, only 0.76 mg/L of the $NH_4^+$-N remained in the effluent of O3, and the nitrogen removal capacity of aerobic denitrification in O1, O2, and O3 accounted for 4.74%, 3.46%, and 3.0% of the total nitrogen removal capacity, respectively. This shows that the contribution of aerobic denitrification to the nitrogen removal of O1 did not differ considerably from that of O2 and O3, indicating that unit A2 fully utilized the influent organic matter with little impact on O1. Aerobic

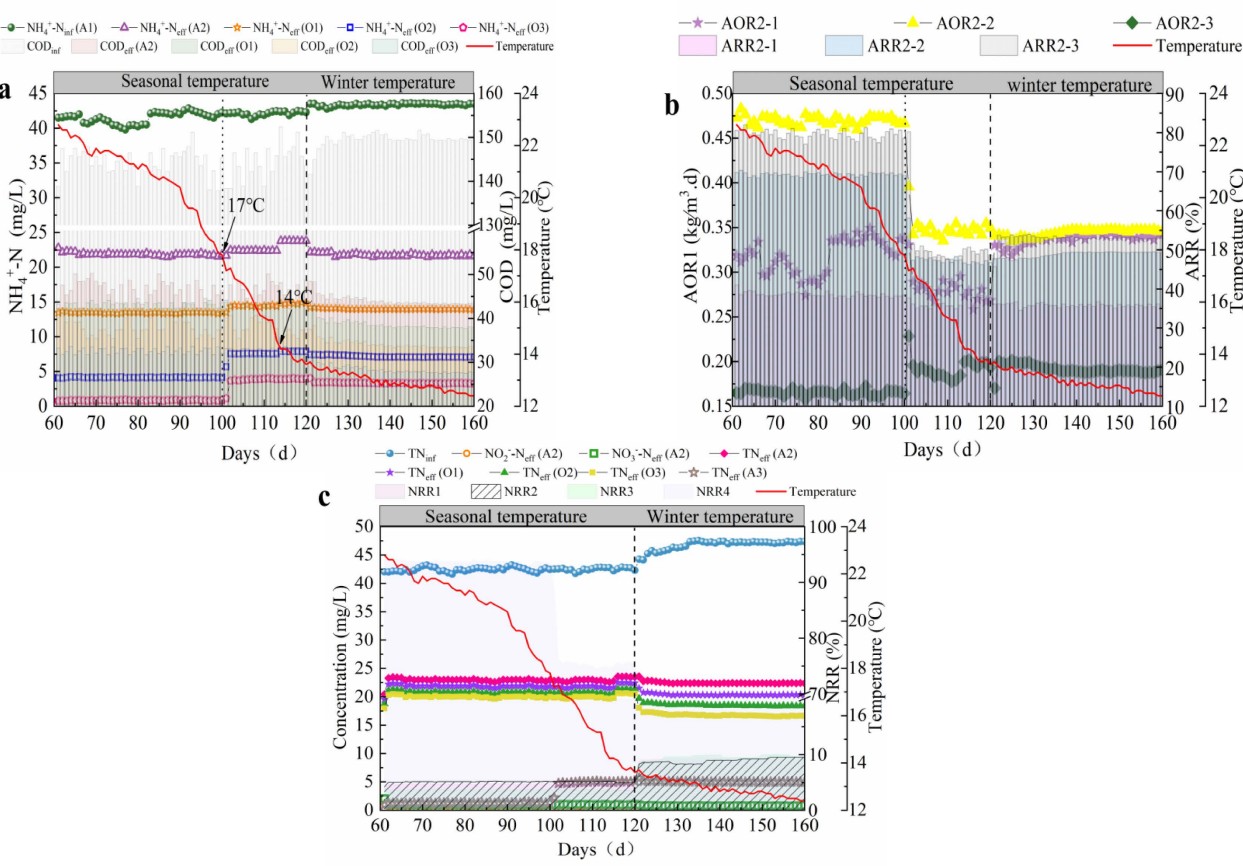

**Fig 5. Performance of the AAOA system wth temperature changes.** (a) changes in the **NH4+-N concentration;** (b) ammonia oxidation rate and removal rate; (c) changes in the TN concentration (AAOA: anaerobic - anoxic - aerobic - anoxic; NH4+-N: ammonia nitrogen; TN: total nitrogen ).

denitrification accounted for 11.2% of the total nitrogen removal of the whole unit. After the A3 unit, the levels of TN and $NH_4^+$-N were 1.89 and 0.26 mg/L, respectively, and the COD concentration was 28.15 mg/L. Unit A3 played a significant role in ensuring that the final discharge met the requirements. Primary and secondary denitrification in the AAOA system accounted for 43.1% and 42.08% of the total nitrogen removal, respectively, and for 85.18% as a whole ([Fig 6]).

In this system, the role of primary denitrification is to fully utilize the organic matter brought by the influent as a carbon source, while the role of secondary denitrification is to ensure the final effluent. Because of this, primary and secondary denitrification are both important to total nitrogen removal. According to Equations 5 to 7, the total amount of nitrogen removed from the AAOA system in the stable period was 235.5 g/d; the total amount of nitrogen removed by aerobic denitrification was 26.37 g/d, while 200.62 g/d was removed by denitrification.

## Impact of organic matter on the performance of nitrification

The impact of organic matter on the performance of nitrification units in the AAOA and AOA systems was explored by a comparison of their performance. The specific operational performance of the AOA system is shown in the supporting materials ([S1 Fig]).

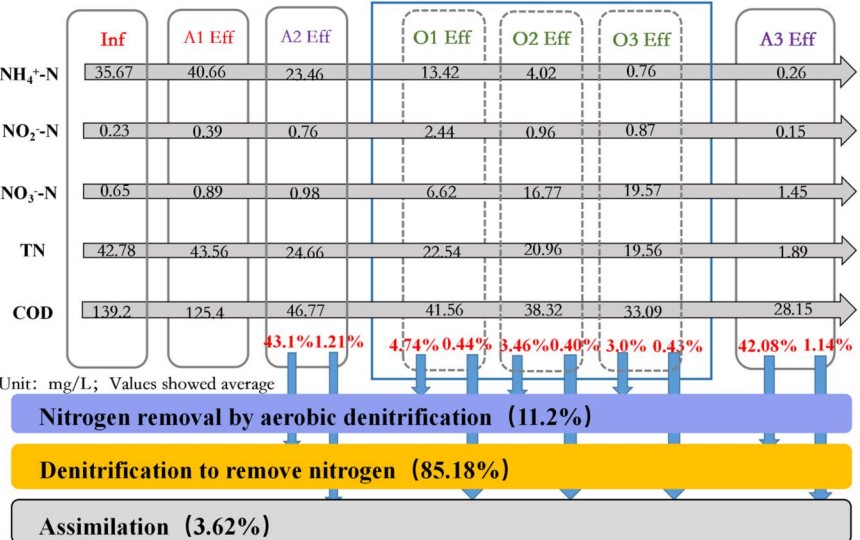

**Fig 6. Nitrogen variation and denitrification contribution of each unit of the AAOA system (AAOA: Anaerobic - anoxic - aerobic - anoxic).**

The stages were > 17 °C, from 17–14 °C, and < 14 °C. The mean values of O1, O2, and O3 in AAOA and AOA during these stages were named either A or F (for AAOA or AOA, respectively);1, 2, or 3 for O1, O2, and O3, respectively, and -1, -2, -3 for each stage, respectively; for example, AAOA O1 at < 14 °C was named A1-3.

The AOA system lacked A2 units, meaning that influent organic matter affects the AOR value of the O1 unit (Fig 7). When the temperature was > 17 °C, the AOR and ARR of the O1 unit were 0.22–0.23 kg/m³·d and 17.5%, whereas those of the O1 unit of the AAOA were 0.32–0.33 kg/m³·d and 38.7%, respectively. As the temperature decreased from 17 °C to 14 °C, the AOR and ARR of the O1 unit in the AOA decreased to 0.17–0.18 kg/m³·d and 12.9%, respectively. When the temperature decreased further to < 14 °C, the levels of AOR and ARR levels decreased again to 0.11–0.12 kg/m³·d and 13.0%, respectively. Compared to the AAOA at low temperatures, the AOR and ARR in the AOA only decreased slightly. The O1 unit of AOA was highly affected by temperature. This indicated that when the influent organic matter entered the nitrification unit, the heterotrophic bacteria competed with the nitrifying bacteria for DO, affecting nitrification efficiency and reducing their ability to resist low temperatures [38]. The ammonia oxidation performance of O2 and O3 in the AOA was significantly improved and reached levels greater than those in the AAOA. This indicated that once the organic matter consumption of the AOA O1 unit was complete, the subsequent fillers still had high activity; this was related to the low O1 performance and high $NH_4^+$-N concentration in the effluent. A comparison of the NRR of the O1 unit between AOA and AAOA showed that there was a significantly larger NRR in the AOA. This was attributed to the use of influent organic matter for denitrification by the aerobic denitrifying bacteria, which resulted in an NRR level of 14.5%. Within the temperature range of 17–14 °C, the NRR still maintained a high level (14.7%). However, when the temperature dropped < 14 °C, the NRR decreased to 5.0%, indicating a decrease in the performance of aerobic denitrifying bacteria.

As shown above, the O1 performance of the AOA system decreased significantly owing to the influence of influent organic matter, and the system was less resistant to low temperatures than the AAOA system. These findings indicate that controlling the amount of organic matter that enters the nitrification stage is also an important factor in maintaining the stability of the nitrification unit.

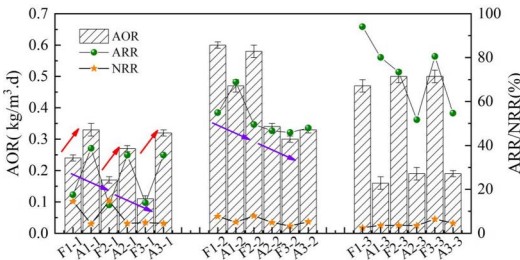

**Fig 7. A comparison of the impact of organic matter on the nitrification performance between the AAOA and AOA systems.**

## Impact of hydrolytic acidification on the performance of the primary denitrification unit

The impact of the A1 unit performance on the A2 unit is shown in Fig 8. After 60 min of hydrolysis and acidification, the concentrations of $NO_2^-$-N and $NO_3^-$-N in the A2 effluent were both lower (0.75 and 0.96 mg/L, respectively), and the TN decreased from 42.1 ± 0.5 mg/L in the influent to 22.56 ± 0.2 mg/L. As the reaction time of A1 decreased (30 min), the concentrations of $NO_2^-$-N and $NO_3^-$-N in the effluent of A2 increased to 3.62 ± 0.2 and 2.1 ± 0.1 mg/L, respectively. This indicated that the amount of simple organic matter remaining after hydrolysis and acidification was no longer sufficient to fuel denitrification, leading to the accumulation of $NO_2^-$-N; when the hydrolysis-acidification reaction ceased and the influent directly entered A2, the $NO_2^-$-N accumulated to 6.11 ± 0.2 mg/L. In addition, the $NO_3^-$-N increased to 3.67 ± 0.1 mg/L. Owing to the lack of hydrolysis and acidification, the biodegradability of the influent could not be increased, which resulted in a decrease in the efficiency of denitrification. Overall, these results indicate that the hydrolysis and acidification stage has a significant impact on the performance of A2. Chang et al (2001) [39] found that the amount of rapidly degradable organic matter in the COD of raw water in urban sewage accounted for 19.19% of the total COD, soluble inert organic matter (non-biodegradable) accounted for 9.74%, suspended inert organic matter (non-biodegradable) accounted for 4.21%, and the active microbial components accounted for 7.27%; the largest proportion was suspended slowly-degradable organic matter, which accounted for 59.59%. However, Pai (2007) [40] found that the $NO_3^-$-N in the A²O anoxic pool decreased from 12.41 to 8.4 mg/L, which indicated a low removal efficiency, possibly caused by the poor hydrolysis and acidification of the A²O anaerobic segment. The results indicate that a hydrolysis-acidification section that performs well can transform the organic matter that is generally resistant to degradation into an available carbon source for denitrification; in contrast, a hydrolysis-acidification section that performs poorly will reduce denitrification efficiency and increase the reaction time. The hydrolysis and acidification performance of the anaerobic section in the common A²O process was poor, which explains the difficulties in improving the nitrogen removal effect of the anoxic section.

## Internal structure of fillers

The internal structures of the nitrification and denitrification fillers were observed by SEM. The electron microscope images of nitrifying fillers were magnified by 2k, 5k, 10k, and 20k, respectively (Fig 9a–d). The overall structure of the embedded nitrification filler was characterized by a tight network structure, with large numbers of bacteria in the gaps (Fig 9(a)(b)). This structure facilitates the adhesion of bacteria and provides space for bacterial growth, which is also the reason why the immobilized filler produces less mud [41,42]. The main types of bacteria in the

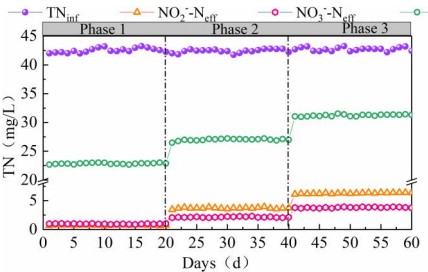

**Fig 8. Effect of hydrolytic acidification on the performance of a primary denitrification unit.**

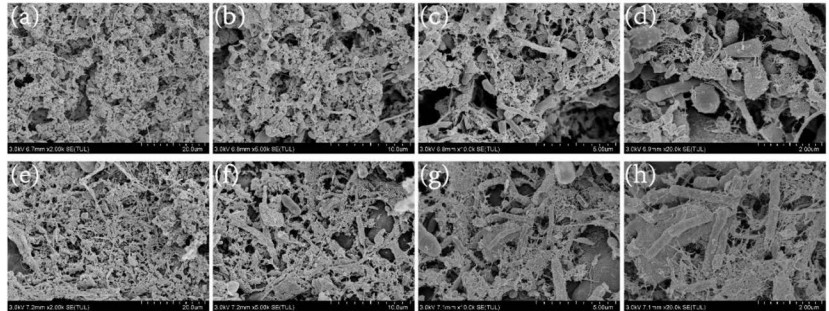

**Fig 9. SEM images of immobilized fillers: (a) nitrification filler magnified by 2k times; (b) nitrification filler magnified by 5k times; (c) nitrification filler magnified by 10k times; (d) nitrification filler magnified by 20k times; (e) denitrification filler magnified by 2k times; (f) denitrification filler magnified by 5k times; (g) denitrification filler magnified by 10k times; (h) denitrification filler magnified by 20k times (SEM: scanning electron microscopy).**

nitrifying filler were ellipsoidal and rod-shaped bacteria (Fig 9(c),(d)). Zhao et al. (2018) [43] found that the nitrite bacteria are primarily spherical. The morphology of the bacterial cells in the figure suggests that *Nitrosococcus* could be found here. The main forms of ammonia-oxidizing bacteria (AOB) are spherical, ellipsoidal, rod-shaped, tightly spiraled, and slender arcs. The main forms of nitrite-oxidizing bacteria (NOB) are spherical, pear-shaped, slender rod-shaped, multi-shaped rod-shaped, and loose spirals. The electron microscopy images of denitrification fillers at different magnifications (Fig 9(e-h)) showed that the internal structure of the also contained numerous bacteria, most of which were rod-shaped (Fig 9(e)). Wong et al. (2005) [44] found that most of the immobilized fillers were occupied by rod-shaped denitrifying bacteria, with a small number of spherical bacteria, which is consistent with the findings of this study.

## Community structure analysis of the AAOA system

### Comparative analysis of the community structure involved in primary and secondary denitrification

Fig 10 shows the abundance of bacteria that were involved in primary and secondary denitrification in the AAOA system to compare the differences in community structure during the long-term operation of denitrification fillers with different carbon sources. The denitrifying bacteria in A2 and A3 primarily included *Arcobacter, Thauera, Comamonas, Thiobacillus, Pseudomonas, Azoarcus, Flavobacterium, Defluviimonas*, and *Thermomonas*, with little difference between the two; however, there were differences in abundance. In A2 and A3, *Arcobacter, Thauera*, and *Comamonas* predominated; the abundance of *Arcobacter* was 14.34 and 21.23%, respectively. *Pseudomonas* and *Thauera* were typical concurrent autotrophic

denitrifying bacteria, while *Arcobacter*, *Thiobacillus*, and *Azoarcus* were the primary autotrophic denitrifying microorganisms. High-throughput sequencing previously identified *Thauera* and *Paracoccus* in the denitrifying immobilized filler as the predominant bacterial genera. The abundance of denitrifying bacteria commonly found in wastewater treatment plants, such as *Pseudomonas* and *Comamonas*, increased significantly with the operation in actual sewage. The denitrification properties of *Flavobacterium, Defluviimonas*, and *Thermomonas* have also been verified, and these three bacteria are more abundant in A3 than A2.

In addition to the dominant denitrifying bacteria in A2 and A3, some bacterial genera capable of hydrolysis-acidification were also found, including *Ornatilinea, Longilinea, Ignavibacterium,* and *Anaerolinea. Ornatilinea* has been shown to ferment sugars and amino acids and then convert them into acetic acid and other volatile fatty acids. *Longilinea* can metabolize a variety of carbohydrates, and Anaerolinea can metabolize a variety of carbohydrates to produce organic acids. The various hydrolysis-acidification strains were more abundant in A2 than in A3; the abundance of *Ornatilinea* was 4.45% and 1.34%, respectively, and that of *Longilinea* was 2.22% and 0.67%. The reason for this is that A2 primarily utilizes the organic matter in raw water as a carbon source; although there is a hydrolysis-acidification unit, owing to fluctuations in actual sewage quality, there may still be some complex organic matter that enters the system, leading to an increase in the abundance of some hydrolysis-acidification bacteria. A3 primarily utilizes sodium acetate as a carbon source, so there were relatively few hydrolytic acidification functional bacteria present. Some common bacterial genera were found at similar levels in these two reactors.

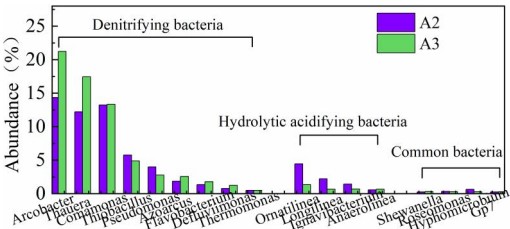

**Fig 10. The bacterial community structure of the denitrification unit before and after the AAOA system.**

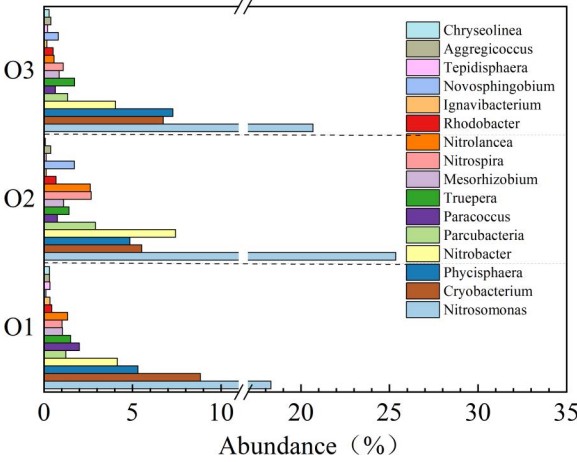

**Fig 11. The bacterial community structure of the nitrification unit in the AAOA system.**

**Analysis of the community structure of nitrification units.** As shown in Fig. 11, the nitrobacteria in O1, O2, and O3 primarily included *Nitrosomonas, Nitrobacter*, and *Nitrospira*. The nitrobacteria in O2 were highly abundant. *Nitrosomonas* was the most abundant (25.36%), followed by *Nitrobacter* (7.44%), *Nitrospira* (2.67%), and *Nitrolancea* (2.62%). There were fewer *Nitrosomonas* in O1 (18.3%) than in O2. Some studies have found that high DO is more conducive to the enrichment of *Nitrosomonas* and *Nitrospira*; this is different from the results in this study, possibly because the small amount of organic matter in the A2 effluent may have affected the abundance of some nitrobacteria. The nitrobacteria in O3 were similarly abundant to those of O1 but lower in abundance than those of O2. This may be because of the high performance of O2, which led to a low concentration of matrix that entered O3, resulting in a decrease in bacterial abundance under the action of long-term low load. As shown in Fig. 11, there were still aerobic denitrifying bacteria. *Paracoccus* was less abundant in O1, O2, and O3, although it was more abundant in O1 than in O2 and O3. Although the AAOA system has an A2 unit to utilize COD in raw water, a small amount of organic matter could enter the nitrification unit, meaning that a small amount of aerobic denitrification could occur. As previously shown, aerobic denitrification bacteria were present in immobilized inoculation sludge. The reason for the reduction in abundance is also related to their small role in the system. *Ignavibacterium* was also less abundant in the system, though it was more abundant in O1 (0.35%) than in O2 and O3. These differences can also be attributed to the influent COD.

## Implications

The AAOA system utilizes immobilized biological fillers in the anaerobic hydrolysis and acidification section. Because it does not require sludge reflux to maintain biomass, it maintains a good anaerobic state to fully promote hydrolysis and acidification. The traditional A$^2$O, MBR process and the combination process with a separate hydrolysis and acidification tank all require sludge reflux, which makes it difficult to maintain an effective anaerobic state. The AAOA system has a primary denitrification stage, meaning that less organic matter enters the nitrification stage. The direct entry of organic matter into the nitrification unit in traditional systems has a greater impact on the performance of nitrifying bacteria; competition between heterotrophic and nitrifying bacteria results in reduced ammonia-nitrogen oxidation rates. However, owing to the poor performance of the anaerobic hydrolysis and acidification stage in the traditional nitrogen removal process, denitrification in the anoxic section is insufficient. Because of this, the organic matter in the aerobic section needs to be removed, which will inevitably affect the performance of nitrifying bacteria.

Compared with the traditional nitrogen removal process, the AAOA system is effective at removing nitrogen because of the setting of the primary denitrification stage of hydrolytic acidification. This ensures the full use of influent organic matter, reducing the concentration that enters the nitrification stage as much as possible. Consequently, there are no limitations on the performance of nitrifying bacteria, resulting in better overall nitrogen removal performance.

## Conclusion

To solve the key core problems facing traditional activated sludge treatment of municipal wastewater, such as low biochemical efficiency and raw organic matter utilization rate, an AAOA nitrogen removal system was constructed and the characteristics and overall performance of each unit were systematically studied. This new system provides an effective method for the removal of nitrogen from municipal wastewater. Compared

with existing biological nitrogen removal processes, AAOA was more effective. When the total HRT of the system was 6.4 h and that of the nitrification unit HRT was 3 h, the effluent $NH_4^+$-N was stabilized at 0.78–0.84 mg/L, and the TN reached 1.5–2 mg/L. When the system R was 90%, the AOR of O1 was increased by 0.33–0.35 kg/m³·d; the effluent COD of A1 was 45.5-48.3 mg/L, and the $NH_4^+$-N remained below 1 mg/L. At low temperatures (<14 °C), the $NH_4^+$-N and TN of the effluent still met the national standard Class A standard.

Organic matter can still affect the performance of the nitrification unit. Although the nitrification filler contains highly efficient nitrifying bacteria, the heterotrophic bacteria in sewage will affect the oxidation of $NH_4^+$-N when there is abundant organic matter. First-stage denitrification is necessary in the AAOA system, which was particularly seen during quarterly temperature changes. Above 17 °C, the AOR and ARR of AOA were 0.22–0.23 kg/m³·d and 17.5%, respectively, while those of the AAOA were 0.32–0.33 kg/m³·d and 8.7%, respectively. When the temperature decreased to 17 °C or lower, the AOR and ARR of AOA decreased to 0.11–0.12 kg/m³·d and 13.0%, respectively, while those of the AAOA only slightly decreased. The change in the organic composition of raw water through altering the HRT of A1 to 60 minutes was crucial to improving denitrification performance and maintaining effluent $NO_2^-$-N and $NO_3^-$-N at a low level.

The AAOA pilot process demonstrates effective biological nitrogen removal and addresses key challenges that the traditional process struggles to overcome, offering valuable insights for process improvement.

## Glossary

| AAOA | Immobilized filler nitrogen removal system (with A2) |
|------|------------------------------------------------------|
| AOA | Immobilized filler nitrogen removal system (no A2) |
| A1 | Hydrolytic-acidification unit |
| A2 | Primary denitrification unit |
| O1 | Nitrification unit 1 |
| O2 | Nitrification unit 2 |
| O3 | Nitrification unit 3 |
| A3 | Secondary denitrification unit |

## Supporting information

**S1 File. This file contains supporting Fig.**
(DOCX)

## Acknowledgements

We are very grateful to the sample testing companies involved in this study for their cooperation in sampling.

## Author contributions

**Data curation:** Xuyan Liu.

**Formal analysis:** Jiawei Wang.

**Funding acquisition:** Xuyan Liu, Hong Yang.

**Software:** Jiawei Wang.

**Writing – original draft:** Xuyan Liu.

**Writing – review & editing:** Hong Yang.

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
