## [Decision Letter · Decision Letter 0]

30 Sep 2024

PONE-D-24-36163

The performance of an immobilized biological filler nitrogen removal system

PLOS ONE

Dear Dr. Wang,

Thank you for submitting your manuscript to PLOS ONE. After careful consideration, we feel that it has merit but does not fully meet PLOS ONE’s publication criteria as it currently stands. Therefore, we invite you to submit a revised version of the manuscript that addresses the points raised during the review process.

We look forward to receiving your revised manuscript.

Kind regards,

Abdelamjeed Adam Lagum

Academic Editor

PLOS ONE

**Reviewers' comments:**

**Reviewer's Responses to Questions**

**Comments to the Author**

1. Is the manuscript technically sound, and do the data support the conclusions?

Reviewer #1: Yes

Reviewer #2: Yes

Reviewer #3: No

2. Has the statistical analysis been performed appropriately and rigorously?

Reviewer #1: N/A

Reviewer #2: No

Reviewer #3: I Don't Know

3. Have the authors made all data underlying the findings in their manuscript fully available?

Reviewer #1: Yes

Reviewer #2: Yes

Reviewer #3: No

4. Is the manuscript presented in an intelligible fashion and written in standard English?

Reviewer #1: No

Reviewer #2: No

Reviewer #3: No

**5. Review Comments to the Author**

**Reviewer #1:**  In this manuscript, the authors conducted a pilot study on nitrogen removal from municipal sewage using immobilized fillers.

I recommend publication only if the following issues can be addressed.

1- Highlight points should be presented.

2- Line 26: AAOA nitrogen removal system. Please use the full term in its first appear. Use “Anaerobic - anoxic - aerobic - anoxic (AAOA)” instead of using abbreviations.

3- Proofreading is essential as there are many mistakes. Please get the language of the study checked by a native English language speaker.

4- The knowledge gap has not been indicated properly in the introduction

5- Novelty of the studied is not well-established.

6- Include a nomenclature section before introduction.

7- You cited "old references" not too many of them were printed after 2017.

8- In order to establish the significance and value of the study, it is necessary to provide a comprehensive rationale for the research, which emphasizes its relevance and unique contributions to the current scholarly discourse. This will strengthen the study's originality and scholarly impact. The following publications related to biological N removal are useful suggested studies to get the benefit to update this part

I. Integrated electro-anammox process for nitrogen removal from wastewater

II. Simultaneous nitrification and denitrification by controlling current density and dissolved oxygen supply in a novel electrically-induced membrane bioreactor

III. Modification of nitrifying microbial community via DC electrical field application

IV. Coupling membrane electro-bioreactor with anammox process to treat wastewater at low temperatures

9- All figures should be optimized

10- The figure captions should be written with more informative.

11- The conclusion should be rewritten. For example, this section “We explored the performance of an AAOA nitrogen removal system constructed by hydrolytic acidification, nitrification, and denitrification immobilized fillers, with the following conclusions: When the total HRT of the system is 6.4 h and the nitrification unit HRT is 3 h, the effluent NH4+-N can be stabilized at 0.78–0.84 mg/L, 551 and the TN can reach 1.5–2 mg/L.” I don't think this section is clear and grammatically accurate.

**Reviewer #2:**  The paper presents a study on the “The performance of an immobilized biological filler nitrogen removal system”. It is a topic of interest to researchers in the related areas since removal of nitrogen from wastewater is an important and continuous objective.

The paper needs some improvement before acceptance for publication.

My detailed comments are as follows:

(1) Abstract:

It is noted that the abstract is very brief and should be enriched to show the problem solved and the novelty of the research.

Moreover, the abstract does not show the methodology used throughout the article.

(2) English and editing:

The paper needs careful editing by someone with expertise in technical English editing who pays particular attention to English grammar, spelling, and sentence structure so that the study's goals, methods, and results are clear to the reader.

(3) Figures:

The figures are too small to read and recognize the results, so please maximize and refine the shown figures.

Please show the comparison and differences between the conventional methods and the paper method for removal of nitrogen. (The paper is poor for showing the literature review and previous work).

(4) Could you please explain how and why you chose the values of the different important parameters?

(5) The equations should be written using and following the journal guidelines.

(6) Please use subscript and superscript while editing “for example O3 not O3”

(7) Line 98, “These not this”.

(8) Line 114, please revise.

(9) Line 319, “decreased not was decreased”.

(10) Line 353, please revise.

(11) How did you get or measure the characteristics of the wastewater?

(12) What is PVA, I suggest you write abbreviations.

(13) Please revise section 2.1.

(14) The innovation and advantages of the method or your work need to be emphasized and summarized whether in Introduction or Methodology.

(15) I can tell that the authors did not follow the Journal guidelines for the structure of the sections and the references.

(16) The references are few compared with the importance of the topic. Please try to double the references at least with more recent references.

Thank you for your effort and I am looking forward receiving the revised version of this good paper.

**Reviewer #3:**  The manuscript describes a new reactor set up to improve nitrogen removal during wastewater treatment. The ideas are interesting and results seem to be show an improvement in nitrogen removal in the novel reactors.

But the data presented does not support all the results and does not make it possible to repeat the experiment. Especially the Materials and Methods section is incomplete.

It does not state how the fillers where made in detail or how and what bacteria where immobilised in them. Similarly the reactors set up and operation is not stated in detail and any dimensions are missing.

The discussion of the results could be much more in depth and next to the SEM images studying the microbial community in the fillers would give a much better understanding the system.

6. PLOS authors have the option to publish the peer review history of their article (what does this mean? ). If published, this will include your full peer review and any attached files.

**Do you want your identity to be public for this peer review?**  For information about this choice, including consent withdrawal, please see our Privacy Policy .

Reviewer #1: No

Reviewer #2: No

Reviewer #3: No

---

## [Author Response · Author response to Decision Letter 0]

31 Oct 2024

Response to Reviewer

Dear Editors and Reviewers:

Thank you for your letter and for the reviewers’ comments concerning our manuscript entitled The performance of an immobilized biological filler nitrogen removal system -- Realizing independent operation at each stage (manuscript number: PONE-D-24-36163). All comments are valuable and very helpful for revising and improving our paper. We have carefully studied the comments and have revised our manuscript accordingly. The main corrections in the manuscript and the responds to the reviewer’s comments are as following:

To Reviewer #1

Thank you very much for pointing out these issues. We have revised our manuscript accordingly.

Comments:

1.Highlight points should be presented.

Thank you for this comment. We add Highlight to the submission as follows: “1. An independent biological nitrogen removal system based on immobilized fillers was proposed; 2. The efficient separation of functional bacteria for denitrification of municipal sewage was realized; 3. The process system can meet the requirements of "Surface water Environmental Quality Standard" (GB-3838-2002) quasi Class IV; 4. An independent hydrolyzing and acidizing device composed of immobilized fillers was constructed with high performance. ”

2. Line 26: AAOA nitrogen removal system. Please use the full term in its first appear. Use “Anaerobic - anoxic - aerobic - anoxic (AAOA)” instead of using abbreviations.

Thank you for this comment; we have revised it.

The revised section now reads as follows:

“Abstract

Immobilized fillers, including hydrolytic-acidification, nitrification and denitrification, were prepared in this study to improve the nitrogen removal performance of the traditional activated sludge process. An anaerobic-anoxic-aerobic anoxic nitrogen removal system was constructed to effectively separate the functional bacteria that remove nitrogen.” Revised in the manuscript Abstract, shown in the underlined section.

3. Proofreading is essential as there are many mistakes. Please get the language of the study checked by a native English language speaker.

Thank you for this comment. It is our negligence and we are sorry about this. We had a native English speaker proofread the manuscript's language.

4. The knowledge gap has not been indicated properly in the introduction.

Thank you for this comment. We have made a major revision to the abstract, please refer to it.

The revised section now reads as follows:

Abstract

Immobilized fillers, including hydrolytic-acidification, nitrification and denitrification, were prepared in this study to improve the nitrogen removal performance of the traditional activated sludge process. An anaerobic-anoxic-aerobic anoxic nitrogen removal system was constructed to effectively separate the functional bacteria that remove nitrogen. The correlation and influence between each unit were explored by studying the operating performance of the system. The levels of ammonia nitrogen and total nitrogen in the effluent could be stabilized at 0.75–0.83 and 1.5–2 mg/L, respectively, when the total hydraulic retention time (HRT) of the system was 6.4 h and the nitrification unit HRT was 3 h. These parameters improved significantly compared with the traditional process of activated sludge. The unit performance study showed that a reduction in the time of hydrolytic-acidification to 0 min resulted in an increase in the levels of nitrite nitrogen and nitrate nitrogen in the effluent of unit A2 to 6.11 ± 0.2 mg/L and 3.67 ± 0.1 mg/L, respectively. Thus, a highly active hydrolysis-acidification stage is the prerequisite for A2 to fully utilize the raw organic matter in the water to remove nitrogen. When the raw organic matter in the water entered the O1 unit directly (without an A2 unit), the ammonia oxidation rate (AOR) decreased significantly (from 0.32–0.33 to 0.22–0.23 kg/m3⋅d). At a low temperature, the AOR of O1 unit decreased even more (to 0.11–0.12 kg/m3⋅d). At this time, the AOR that was not influenced by organic matter only decreased slightly. This indicates that causing the organic matter to enter the nitrification stage is essential to maintain its stability and resist low temperatures. Revised in the manuscript Abstract, shown in the underlined section.

5.Novelty of the studied is not well-established

Thank you for this comment. The anaerobic- Anoxic-aerobic anoxic (AAOA) nitrogen removal system established in this manuscript is innovative for the following reasons:

Traditional activated sludge process is one of the most widely used processes in urban sewage treatment plants due to its wide application range and mature technology. However, due to the operation mode of the single sludge system, various functional bacteria (hydrolytic acidification, nitrification, denitrification removal bacteria) with different physiological characteristics and generation cycle exist in the same system for reaction, which makes it difficult to give full play to their respective advantages, and makes it difficult to improve the overall performance of the process (Zanetti et al., 2017; Wang et al., 2014). This results in many problems, such as low biochemical efficiency, poor stability, low utilization rate of organic matter in raw water and unable to give full play to biological nitrogen and phosphorus removal efficiency. Therefore, the development of nitrogen removal systems independently carried out by various functional bacteria has become a new trend of municipal sewage treatment.

In this study, the hydrolytic-acidification unit with stable anaerobic state was established by using hydrolytic-acidification immobilized filler (Wang et al., 2021). The method not only stores a high concentration of active biomass, but also provides a stable environment for the growth of hydrolyzed acidifying bacteria. On the basis of improving the insufficient activated sludge process, hydrolytic-acidification can be fully utilized without affecting the design of the subsequent process, aiming to provide a more stable and effective new method for the full utilization of organic matter in the raw water of municipal sewage and TN removal. Then, the AAOA nitrogen removal system was established by hydrolytic-acidification, nitrification and denitrification immobilized fillers to achieve effective separation of various functional bacteria. The performance of the system was analyzed, studied and improved through process regulation, providing an effective method for efficient biological nitrogen removal of municipal sewage.

Reference

[1]Zanetti L, Frison N, Nota E, et al. Progress in real-time control applied to biological nitrogen removal from wastewater. A short-review[J]. Desalination, 2012, 286(2012): 1-7.

[2]Wang Y, Li W G, Irini A, et al. Removal of organic pollutants in tannery wastewater from wet-blue fur processing by integrated Anoxic / Oxic (A/O) and Fenton: Process optimization[J]. Chemical Engineering Journal, 2014, 252: 22-29.

[3] Wang J, Fan Y C, Chen Y P. Nitrogen removal performance and characteristics of gel beads immobilized anammox bacteria under different PVA: SA ratios[J]. Water environment research, 2021, 93(9): 1627-1639.

6.Include a nomenclature section before introduction.

Thank you for this comment. We added a nomenclature section before the introduction.

The revised section now reads as follows:

The list of self-named abbreviations

AAOA Immobilized filler nitrogen removal system (with A2)

AOA Immobilized filler nitrogen removal system (no A2)

A1 Hydrolytic-acidification unit

A2 Primary denitrification unit

O Nitrification unit

A3 Secondary denitrification unit

Common Abbreviations

TN Total Nitrogen

NH4+ -N Ammonia Nitrogen

NO2--N Nitrite Nitrogen

NO3--N Nitrate Nitrogen

COD Chemical Oxygen Demand

DON Dissolved Organic Nitrogen

DO Dissolved Oxygen

HRT Hydraulic Retention Time

AOR Ammonia Oxidation Rate

ARR Ammonia Nitrogen Removal Rate

NRR TN Removal Rate

SEM Scanning electron microscopy

VFAs Volatile Fatty Acids

7. You cited "old references" not too many of them were printed after 2017.

Thank you for this comment. We have updated the references in the manuscript .

8.In order to establish the significance and value of the study, it is necessary to provide a comprehensive rationale for the research, which emphasizes its relevance and unique contributions to the current scholarly discourse. This will strengthen the study's originality and scholarly impact. The following publications related to biological N removal are useful suggested studies to get the benefit to update this part

I. Integrated electro-anammox process for nitrogen removal from wastewater

II. Simultaneous nitrification and denitrification by controlling current density and dissolved oxygen supply in a novel electrically-induced membrane bioreactor

III. Modification of nitrifying microbial community via DC electrical field application

IV. Coupling membrane electro-bioreactor with anammox process to treat wastewater at low temperatures

Thank you for this comment. We think the references you provided are very valuable and we have cited them in the manuscript according to the content.

9. All figures should be optimized.

Thank you for this comment. We have optimized pixel and font sizes for all figures.

10.The figure captions should be written with more informative.

Thank you for this comment. We have optimized the figure captions.

The revised section now reads as follows:

Fig 1. Schematic diagram of the Anaerobic - anoxic - aerobic - anoxic (AAOA) nitrogen removal process.

Fig. 2. Schematic diagram of the process changes under the influence of organic matter, including the AOA and AAOA nitrogen removal processes.

Fig 3. AAOA system optimized reflux ratio stage performance: (a) Changes in NH4+-N; (b) AOR and ARR; (c) Changes in TN ( (AAOA: Anaerobic - anoxic - aerobic - anoxic; NH4+-N: Ammonia nitrogen; AOR: Ammonia oxidation rate; ARR: Ammonia removal rate).

Fig. 4. Optimization of the HRT stage performance in the AAOA system: (a) Changes in NH4+-N concentration; (b) AOR and ARR;(c) Changes in TN (AAOA: Anaerobic - anoxic - aerobic - anoxic; HRT: Hydraulic Retention Time; NH4+-N: Ammonia nitrogen; AOR: Ammonia oxidation rate; ARR: Ammonia removal rate).

Fig 5. Performance of the AAOA system at different stages of temperature changes: (a) Changes in the NH4+-N concentration; (b) Ammonia oxidation rate and removal rate; (c) Changes in the TN concentration (AAOA: Anaerobic - anoxic - aerobic - anoxic; NH4+-N: Ammonia nitrogen; TN: Total nitrogen).

Fig 6. Nitrogen variation and denitrification contribution of each unit of the AAOA system (AAOA:Anaerobic - anoxic - aerobic - anoxic).

Fig 7. A comparison of the impact of organic matter on the nitrification performance between the AAOA and AOA systems.

Fig 8. Effect of hydrolytic acidification on the performance of a primary denitrification unit.

Fig 9. SEM images of immobilized fillers: (a) nitrification filler magnified by 2k times; (b) nitrification filler magnified by 5 k times; (c) nitrification filler magnified by 10 k times; (d) nitrification filler magnified by 20 k times; (e) denitrification filler magnified by 2 k times; (f) denitrification filler magnified by 5 k times; (g) denitrification filler magnified by 10 k times; (h) denitrification filler magnified by 20 k times (SEM: Scanning Electron Microscopy).

Fig 10. The bacterial community structure of the denitrification unit before and after the AAOA system.

Fig 11. The bacterial community structure of the nitrification unit in the AAOA system.

11. The conclusion should be rewritten. For example, this section “We explored the performance of an AAOA nitrogen removal system constructed by hydrolytic acidification, nitrification, and denitrification immobilized fillers, with the following conclusions: When the total HRT of the system is 6.4 h and the nitrification unit HRT is 3 h, the effluent NH4+-N can be stabilized at 0.78–0.84 mg/L, 551 and the TN can reach 1.5–2 mg/L.” I don't think this section is clear and grammatically accurate.

Thank you for this comment. We have rewritten the conclusion.

The revised section now reads as follows:

Conclusion

This study examined the performance of an AAOA nitrogen removal system constructed by hydrolytic acidification, nitrification and denitrification immobilized fillers. The overall operation of the system under the background of municipal sewage was studied, including the effects of reflux ratio, HRT and seasonal temperature on the operation of the system. The following conclusions were drawn:

1.Compared with the other existing biological nitrogen removal processes, AAOA removed nitrogen more effectively. When the total HRT of the system is 6.4 h and the nitrification unit HRT is 3 h, the effluent NH4+-N can be stabilized at 0.78–0.84 mg/L. The TN can reach 1.5–2 mg/L. When the system R is 90%, the AOR of O1 can be increased by 0.33–0.35 kg/m3⋅d; the effluent COD of A1 is 45.5-48.3 mg/L, and the NH4+-N can be kept below 1 mg/L. At low temperatures in the winter (<14 ℃), the NH4+-N and TN of the effluent of AAOA system can still reach the national standard Class A standard.

2.Organic matter can still affect the performance of the nitrification unit. Although the nitrification filler contains highly efficient nitrifying bacteria, the heterotrophic bacteria in sewage will affect the oxidation of NH4+-N when there is abundant organic matter. Therefore, first-stage denitrification is necessary in the AAOA system, which is particularly reflected during the process of quarterly temperature changes. Above 17 ℃, the AOR and ARR of AOA were 0.22–0.23 kg/m3⋅d and 17.5%, respectively, while the AOR and ARR of the AAOA were 0.32–0.33 kg/m3⋅d and 8.7%, respectively. When the temperature decreased to 17 ℃ or below, the AOR and ARR of AOA decreased to 0.11–0.12 kg/m3⋅d and 13.0%, respectively, while the AAOA only slightly decreased.

3.The change in the organic composition of raw water through the experimental results of HRT change of A1 is crucial to improving the performance of denitrification. Changing the HRT of the A1 (0, 30, 60 min) to an HRT of 60 min results in effective, A2, and the effluent NO2--N and NO3--N concentrations are maintained at a low level.

Special thanks to you for all your valuable comments!

To Reviewer #2

Thank you very much for your comments and suggestions. We have modified and explained your question.

Comments:

1.Abstract:It is noted that the abstract is very brief and should be enriched to show the problem solved and the novelty of the research. Moreover, the abstract does not show the methodology used throughout the article.

Thank you for this comment. We have rewritten the Abstract and highlighted it with underline and yellow.

The revised section now reads as follows:

Abstract

Immobilized fillers, including hydrolytic-acidification, nitrification and denitrification, were prepared in this study to improve the nitrogen removal performance of the traditional activated sludge process. An anaerobic-anoxic-aerobic anoxic nitrogen removal system was constructed to effectively separate the functional bacteria that remove nitrogen. The correlation and influence between each unit were explored by studying the operating performance of the system. The levels of ammonia nitrogen and total nitrogen in the effluent could be stabilized at 0.75–0.83 and 1.5–2 mg/L, respectively, when the total hydraulic retention time (HRT) of the system was 6.4 h and the nitrification unit HRT was 3 h. These parameters improved significantly compared with the traditional process of activated sludge. The unit performance study showed that a reduction in the time of hydrolytic-acidification to 0 min resulted in an increase in the levels of nitrite nitrogen and nitrate nitrogen in the effluent of unit A2 to 6.11 ± 0.2 mg/L and 3.67 ± 0.1 mg/L, respectively. Thus, a highly active hydrolysis-acidification stage is the prerequisite for A2 to fully utilize the raw organic matter in the water to remove nitrogen. When the raw organic matter in the water entered the O1 unit directly (without an A2 unit), the ammonia oxidation rate (AOR) decreased significantly (from 0.32–0.33 to 0.22–0.23 kg/m3⋅d). At a low temperatur

---

## [Decision Letter · Decision Letter 1]

18 Nov 2024

PONE-D-24-36163R1

The performance of an immobilized biological filler nitrogen removal system -- Realizing independent operation at each stage

PLOS ONE

Dear Dr. Wang,

Thank you for submitting your manuscript to PLOS ONE. After careful consideration, we feel that it has merit but does not fully meet PLOS ONE’s publication criteria as it currently stands. Therefore, we invite you to submit a revised version of the manuscript that addresses the points raised during the review process.

We look forward to receiving your revised manuscript.

Kind regards,

Abdelamjeed Adam Lagum

Academic Editor

PLOS ONE

Journal Requirements:

Additional Editor Comments:

Thank you for your revision of the manuscript. While it has improved significantly, I kindly request improvement of the English language and conducting another review of the equations

Please consider enhancing the title, as an engaging and impactful title can significantly improve the potential for citations. Think about this  for example “Optimizing Nitrogen Removal with an Immobilized Biological Filler System: Realizing Stage-Independent Operational Process “

Reviewers' comments:

Reviewer's Responses to Questions

**Comments to the Author**

1. If the authors have adequately addressed your comments raised in a previous round of review and you feel that this manuscript is now acceptable for publication, you may indicate that here to bypass the “Comments to the Author” section, enter your conflict of interest statement in the “Confidential to Editor” section, and submit your "Accept" recommendation.

Reviewer #1: All comments have been addressed

Reviewer #2: (No Response)

2. Is the manuscript technically sound, and do the data support the conclusions?

Reviewer #1: Yes

Reviewer #2: Yes

3. Has the statistical analysis been performed appropriately and rigorously? 

Reviewer #1: N/A

Reviewer #2: Yes

4. Have the authors made all data underlying the findings in their manuscript fully available?

Reviewer #1: Yes

Reviewer #2: Yes

5. Is the manuscript presented in an intelligible fashion and written in standard English?

Reviewer #1: Yes

Reviewer #2: No

6. Review Comments to the Author

Reviewer #1: (No Response)

Reviewer #2: Dear Authors,

Thank you so much for revising the manuscript in a very good way. However, I'm still willing that you polish the abstract and the conclusion to show the novelty and impact of your results in wastewater treatment major.

Another revision of the equations and English language should be conducted.

Thank you

7. PLOS authors have the option to publish the peer review history of their article (what does this mean? ). If published, this will include your full peer review and any attached files.

**Do you want your identity to be public for this peer review?** For information about this choice, including consent withdrawal, please see our Privacy Policy .

Reviewer #1: No

Reviewer #2: No

---

## [Author Response · Author response to Decision Letter 1]

28 Nov 2024

Response to Reviewer

Dear Editors and Reviewers:

Thank you for your letter and for the reviewers’ comments concerning our manuscript entitled The performance of an immobilized biological filler nitrogen removal system -- Realizing independent operation at each stage (manuscript number: PONE-D-24-36163). All comments are valuable and very helpful for revising and improving our paper. We have carefully studied the comments and have revised our manuscript accordingly. The main corrections in the manuscript and the responds to the reviewer’s comments are as following:

Journal Requirements:

1.Please review your reference list to ensure that it is complete and correct. If you have cited papers that have been retracted, please include the rationale for doing so in the manuscript text, or remove these references and replace them with relevant current references. Any changes to the reference list should be mentioned in the rebuttal letter that accompanies your revised manuscript. If you need to cite a retracted article, indicate the article’s retracted status in the References list and also include a citation and full reference for the retraction notice.

In the first revision, we updated and added the references according to reviewer 2's comments“16.The references are few compared with the importance of the topic. Please try to double the references at least with more recent references”. The underlined section below adds an updated list of references, and we do not cite retracted papers. References marked in yellow and underlined are new additions.

Reference

1.Hisashi S, Hideki O, Bian R, Kamo J, Okabe S, Fukushi K. Macro scale and micro scale analyses of nitrification and denitrification in biofilms attached on membrane aerated biofilm reactors. Water Research. 2004; 38:1633-1641. doi: 10.1016/ j.watres. 2003. 12.020.

2.Shi K, Liang B, Cheng HY, Wang HC, Liu WZ, Li ZL, et al. Regulating microbial redox reactions towards enhanced removal of refractory organic nitrogen from wastewater. Water research. 2024; 258: 121778.1-121778.16. doi: 10.1016/j.watres.2024.121778.

3.Mishra S, Singh V, Cheng L, Hussain A, Ormeci B. Nitrogen removal from wastewater: A comprehensive review of biological nitrogen removal processes, critical operation parameters and bioreactor design. Journal of Environmental Chemical Engineering. 2023; 10(3): 107387. doi: 10.1016/j.jece.2022.107387.

4.Roots P, Sabba F, Wells G. Integrated shortcut nitrogen and biological phosphorus removal from mainstream wastewater: process operation and modeling. Environmental science: Water research & technology. 2020; 6(3): 566-580. doi: 10.1039/c9ew00550a.

5.Zanetti L, Frison N, Nota E, et al. Progress in real-time control applied to biological nitrogen removal from wastewater. A short-review. Desalination. 2012; 286:1-7. doi: 10.1016/j.desal.2011.11.056.

6. Lagum AA. Integrated electro-anammox process for nitrogen removal from wastewater. International Journal of Environmental Science and Technology. 2023; 20: 13061-13072. doi: 10.1007/s13762-023-04839-3.

7. Zhang XY, Liu Y. Circular economy-driven ammonium recovery from municipal wastewater: State of the art, challenges and solutions forward. Bioresource Technology. 2021; 334: 125231. doi: 10.1016/j.biortech.2021.125231.

8. Cruddas PH, Asproulis N, Antoniadis A, Best D, Collins E, Porca E, et al. The impact of hydraulic retention time on the performance of two configurations of anaerobic pond for municipal sewage treatment. Environmental Technology. 2022; 43(25): 3905-3918. doi: 10.1080/09593330.2021.1937331.

9. Pasalari H, Gholami M, Rezaee A, et al. Perspectives on microbial community in anaerobic digestion with emphasis on environmental parameters: A systematic review. Chemosphere. 2020; 270:128618. doi: 10.1016/j.chemosphere. 2020. 128618.

10. Moretto G, Ardolino F, Piasentin A, Girotto L, Cecchi F. Integrated anaerobic codigestion system for the organic fraction of municipal solid waste and sewage sludge treatment: an Italian case study. Journal of Chemical Technology & Biotechnology. 2020; 95: 418-426. doi: 10.1002/jctb.5993.

11. Oehmen A, Zeng RJ, Yuan Z, et al. Anaerobic metabolism of propionate by polyphosphate - accumulating organisms in enhanced biological phosphorus removal systems. Biotech Bioeng. 2005; 91: 43–53. doi: 10.1002/bit.20480.

12. Cheng C, Zhou Z, Qiu Z, et al. Enhancement of sludge reduction by ultrasonic pretreatment and packing carriers in the anaerobic side-stream reactor: Performance, sludge characteristics and microbial community structure. Bioresource Technol. 2018; 249: 298-306. doi: 10.1016/j.biortech.2017.10.043.

13. Zhang YD, Zhang L, Guo B, Zhou Y, Gao MJ, Sharaf A, et al. Granular activated carbon stimulated microbial physiological changes for enhanced anaerobic digestion of municipal sewage. Chemical Engineering Journal. 2020; 400: 125838. doi: 10.1016/j.cej.2020.125838.

14. Xie ZF, Wang ZW, Wang QY, et al. An anaerobic dynamic membrane bioreactor (AnDMBR) for landfill leachate treatment: performance and microbial community identification. Bioresource Technology. 2014; 161: 29-39. doi: 10.1016/j.biortech.2014.03.014.

15. Rostron WM, Stuckey DC, Young AA. Nitrification of high strength ammonia wastewaters: comparative study of immobilisation media. Water Research. 2002; 35: 1169-1178. doi: 10.1016/S0043-1354(00)00365-1.

16. San SP, Adnan R, Ng SL. Statistical optimization of immobilization of activated sludge in PVA/alginate cryogel beads using response surface methodology for p-nitrophenol biodegradation. Journal of Water Process Engineering. 2021; 39: 101725. doi: 10.1016/j.jwpe.2020.101725.

17. Magri A, Vanotti MB, Szoegi AA. Anammox sludge immobilized in polyvinyl alcohol (PVA) cryogel carriers. Bioresource Technology. 2012; 114: 231-240. doi: 10.1016/j.biortech.2012.03.077.

18. Wang XT, Yang H, Zhou YK, Liu XY. Performance and mechanism analysis of gel immobilized anammox bacteria in treating different proportions of domestic wastewater: a valid alternative to granular sludge. Bioresource Technology. 2022; 347: 126623. doi:10.1016/j.biortech.2021.126623.

19. Tringe SG, Rubin EM. Metagenomics: DNA sequencing of environmental samples. Nature Reviews Genetics. 2005; 6: 805-814. doi: 10.1038/nrg1709.

20. Pandi GJ, Raja K, Vijayan V, Sudhagar S. Investigation on the mechanical, water absorption, and tribological performance of calotropis gigantea and abaca fiber reinforced epoxy composites. Journal of Polymer Research. 2024; 31(308). doi: 10.1007/s10965-024-04157-3.

21. Zhang J, Zhou J, Han Y, et al. Start-up and bacterial communities of single-stage nitrogen removal using anammox and partial nitritation (SNAP) for treatment of high strength ammonia wastewater. Bioresource Technology. 2014; 169: 652-657. doi: 10.1016/j.biortech.2014.07.042.

22.Tipaldi CF, Vitols K, Kokis T, Trausa A, Sarakovskis A. Experimental and Theoretical Comparison and Analysis of Surface-Enhanced Raman Scattering Substrates with Different Morphologies. Appl. Sci. 2024; 14(19): 9040. doi: 10.3390/ app14199040.

23.Sallis PJ, Uyanik S. Granule development in a split-feed anaerobic baffled reactor (SFABR). Bioresource Technology. 2003; 89: 255-265. doi: 10.1016/S0960-8524 (03)00071-3.

24.Baloch MI, Akunna JC, Kierans M, et al. Structural analysis of anaerobic granules in a phase separated reactor by electron microscopy. Bioresource Technology. 2008; 99: 922-929. doi: 10.1016/j.biortech.2007.03.016.

25.Lei L, Yao JC, Liu YD, Li W. Performance, sludge characteristics and microbial community in a salt-tolerant aerobic granular SBR by seeding anaerobic granular sludge. International Biodeterioration & Biodegradation. 2021; 163: 105258. doi: 10.1016/j.ibiod.2021.105258.

26.Lagum AA, Elektorowicz M. Modification of nitrifying microbial community via DC electrical field application. Journal of Environmental Chemical Engineering. 2022; 10(3): 107743. doi: 10.1016/j.jece.2022.107743.

27.Gao HX, Fela S, Lin YM, Van Lier JB, Kreuk MD. Structural extracellular polymeric substances determine the difference in digestibility between waste activated sludge and aerobic granules. Water Research. 2020; 181: 115924. doi: 10.1016/j.watres.2020.115924.

28.Castellano-Hinojosa A, Armato C, Pozo C, et al. New concepts in anaerobic digestion processes: recent advances and biological aspects. Appl. Microbiol. Biotechnol. 2018; 102: 5065-5076. doi: 10.1007/s00253-018-9039-9.

29.Coats ER, Appel FJ, Guho N, Brinkman CK, Mellin J. Interrogating the performance and microbial ecology of an enhanced biological phosphorus removal/post-anoxic denitrification process at bench and pilot scales. Water. 2023; 95(4): e10852. doi: 10.1002/wer.10852.

30.Wang C, Liu C, Si X, et al. Study on the Choice of Wastewater Treatment Process Based on the Emergy Theory. Processes. 2021; 9: 1648-1648. doi: 10.3390/ pr9091648.

31.Carlson AL, He H, Yang C, Daigger GT. Comparison of hybrid membrane aerated biofilm reactor (MABR)/suspended growth and conventional biological nutrient removal processes.Water Science and Technology. 2021; 84(14): 1418-1428. doi: 10.2166/wst.2021.062.

32. Zhang Q, Chen W, Yuan CB, et al. A novel pure biofilm system based on aerobic denitrification for nitrate wastewater treatment: Exploring the feasibility of high total nitrogen removal under low-carbon condition. Chemical Engineering Journal. 2023; 480: 147978. doi: 10.1016/j.cej.2023.147978.

33. Baeza JA, Gabriel D, Lafuente J. Effect of internal recycle on the nitrogen removal efficiency of an anaerobic/anoxic/oxic (A2/O) wastewater treatment plant (WWTP). Process Biochemistry. 2004; 39: 1615-1624. doi: 10.1016/S0032 -9592 (03)00300-5.

34. Lagum AA. Simultaneous nitrification and denitrification by controlling current density and dissolved oxygen supply in a novel electrically-induced membrane bioreactor. Journal of Environmental Management. 2022; 322: 116131. doi: 10.1016/j.jenvman.2022.116131.

35. Liu SL, Chen YS, Xiao L. Metagenomic insights into mixotrophic denitrification facilitated nitrogen removal in a full-scale A2/O wastewater treatment plant. Plos one. 2021; 16(4): e0250283. doi: 10.1371/journal.pone.0250283.

36.Wang F, Yang KL, Jiang WQ. Factors affecting the enrichment of heterotrophic nitrifying aerobic denitrifying bacteria in municipal wastewater treatment systems and analysis of differences in nitrogen metabolism. Journal of Water Process Engineering. 2024; 58: 104807. doi: 10.1016/j.jwpe.2024.104807.

37. Lagum AA, Ghriybah MA, Al-Ma'abreh AM. Coupling membrane electro-bioreactor with anammox process to treat wastewater at low temperatures. Arabian Journal of Chemistry. 2023; 16(10): 105165. doi: 10.1016/j.arabjc.2023.105165.

38. Masoudi SMA, Moghaddam AH, Sargolzaei J. Investigation and optimization of the SND–SBR system for organic matter and ammonium nitrogen removal using the central composite design. Environmental Progress & Sustainable Energy. 2017; 37: 1638-1646. doi: 10.1002/ep.12847.

39. Chang EE, Chiang PC, Ko YW, et al. Characteristics of organic precursors and their relationship with disinfection by-products. Chemosphere. 2001; 44(5): 1231-1236. doi: 10.1016/s0045-6535(00)00499-9.

40. Pai TY. Modeling nitrite and nitrate variations in A2O process under different return oxic mixed liquid using an extended model. Process Biochemistry. 2007; 42: 978-987. doi: 10.1016/j.procbio.2007.03.006.

41. Crocetti GR, Bond PL, Banfield JF, et al. Glycogen-accumulating organisms in laboratory-scale and full-scale wastewater treatment processes. Microbiology. 2002; 148: 3353-3364. doi: 10.1099/00221287-148-11-3353.

42. Wang XX, Zhao J, Yu DS, et al. Stable nitrite accumulation and phosphorous removal from nitrate and municipal wastewaters in a combined process of endogenous partial denitrification and denitrifying phosphorus removal (EPDPR). Chemical Engineering Journal. 2019; 355: 560-571. doi: 10.1016/j.cej. 2018.08.165.

43. Zhao WH, Huang Y, Wang MX, et al. Post-endogenous denitrification and phosphorus removal in an alternating anaerobic/oxic/anoxic (AOA) system treating low carbon/nitrogen (C/N) domestic wastewater. Chemical Engineering Journal. 2018; 339: 450 - 458. doi: 10.1016/j.cej.2020.124812.

44. Wong MT, Mino T, Seviour RJ, et al. In situ identification and characterization of the microbial community structure of full-scale enhanced biological phosphorous removal plants in Japan. Water Research. 2005; 39(13): 2901-2914. doi: 10.1016/j. watres.2005.05.015.

Additional Editor Comments:

1.Thank you for your revision of the manuscript. While it has improved significantly, I kindly request improvement of the English language and conducting another review of the equations. Please consider enhancing the title, as an engaging and impactful title can significantly improve the potential for citations. Think about this for example “Optimizing Nitrogen Removal with an Immobilized Biological Filler System: Realizing Stage-Independent Operational Process”

Thank you very much for your comments, we have polished the English language and conducted another review of the equations. At the same time, the title you suggested is very good and we will adopt it, which is “Optimizing Nitrogen Removal with an Immobilized Biological Filler System: Realizing Stage-Independent Operational Process”.

Equations and calculations

The ammonia oxidation rate (AOR) and the ammonia nitrogen removal rate (ARR) were calculated using Equations 1, 2, and 3:

(1)

(2)

(3)

where (NH4+-N)inf is the influent NH4+-N concentration; (NH4+-N)eff is the influent NH4+-N concentration, and tHRT is the hydraulic retention time.

The TN removal rate (NRR) was calculated using Equation 4:

(4)

Where (TN)inf is the influent TN concentration, and (TN)eff is the influent TN concentration.

The distribution of TN in each part of the process was calculated as follows: (5)

The daily N inflow was calculated from the inflow and the TN concentration, including DON, NH4+-N , and NO3--N; the NO2--N concentration was negligible.

(6)

In the AAOA system, the N was removed through primary and secondary denitrification, and the TN amount in this stage was Mass TNA2,A3. Aerobic denitrification occurred in the nitrification stage, where TN was Mass TNo. Since the N removal system was composed of immobilized fillers and did not consider the residual sludge problem, the residual TN was taken as the final effluent N content. The formula for this calculation is as follows:

(7)

The daily effluent N level was calculated from the effluent flow and the TN concentration.

To Reviewer #2

Thank you very much for pointing out these issues. We have revised our manuscript accordingly.

Comments:

1.Thank you so much for revising the manuscript in a very good way. However, I'm still willing that you polish the abstract and the conclusion to show the novelty and impact of your results in wastewater treatment major. Another revision of the equations and English language should be conducted.Thank you

Thank you for this comment. We have revised the abstract and conclusion again, and the specific changes are as follows:

Abstract

Due to the operation mode of traditional activated sludge systems, it is difficult for various functional bacteria to exert their respective advantages. In this study, immobilized fillers for hydrolytic acidification, nitrification, and denitrification were developed to allow independent operation at each stage, enhancing nitrogen removal performance of overall process. The results showed that ammonia nitrogen and total nitrogen levels in the effluent stabilized at 0.75–0.83 and 1.5–2 mg/L, respectively, when the total hydraulic retention time (HRT) of the system was 6.4 h and the nitrification unit HRT was 3 h. These values represented significant improvements compared with the traditional activated sludge process. Unit performance tests revealed that reducing the hydrolytic-acidification time to 0 min increased nitrite nitrogen and nitrate nitrogen levels in the effluent of unit A2 to 6.11 ± 0.2 mg/L and 3.67 ± 0.1 mg/L, respectively. This demonstrates that an active hydrolysis - acidification

---

## [Editor Report · Decision Letter 2]

3 Dec 2024

Optimizing Nitrogen Removal with an Immobilized Biological Filler System: Realizing Stage-Independent Operational Process

PONE-D-24-36163R2

Dear Dr. Wang,

We’re pleased to inform you that your manuscript has been judged scientifically suitable for publication and will be formally accepted for publication once it meets all outstanding technical requirements.

Kind regards,

Abdelamjeed Adam Lagum

Academic Editor

PLOS ONE
---

## [Editor Report · Acceptance letter]

PONE-D-24-36163R2

PLOS ONE

Dear Dr. Wang,

I'm pleased to inform you that your manuscript has been deemed suitable for publication in PLOS ONE. Congratulations! Your manuscript is now being handed over to our production team.

Kind regards,

on behalf of

Dr. Abdelamjeed Adam Lagum

Academic Editor

PLOS ONE